# The Effects of a Simulation-Based Patient Safety Education Program on Compliance with Patient Safety, Perception of Patient Safety Culture, and Educational Satisfaction of Operating Room Nurses

**DOI:** 10.3390/healthcare11212824

**Published:** 2023-10-25

**Authors:** OkBun Park, MiYang Jeon, MiSeon Kim, ByeolAh Kim, HyeonCheol Jeong

**Affiliations:** 1Department of Nursing, Konkuk University Hospital, Seoul 05030, Republic of Korea; bada@kuh.ac.kr (O.P.); 20080048@kuh.ac.kr (B.K.); 2College of Nursing, Institute of Medical Science, Gyeongsang National University, Jinju 52725, Republic of Korea; myjeon68@gnu.ac.kr; 3College of Nursing, Sahmyook University, Seoul 01795, Republic of Korea; aurora009@hanmail.net

**Keywords:** simulation, operating room, nurse, safety, education

## Abstract

Background: Operating rooms (ORs) are healthcare areas that are high-risk regarding patient safety (PS). The prevention of PS errors such as wrong-site surgery, medication errors, and patient falls is important in the OR. Causes such as having insufficient information, not taking enough care and precautions, and inattention may lead to errors. Ensuring PS in an organization depends on the composition of a PS culture. Method: This study, as equivalent-control-group pretest–posttest research, aimed to develop and apply a simulation-based patient safety education program for operating room nurses and then to examine the influence of the program on patient safety management and compliance. Participants included a total of 45 operating room nurses: 22 in the experimental group and 23 in the control group. In the program, each member of the experimental and control groups underwent two 60 min sessions for a total of 120 min. Person-to-person individual lectures and simulation-based practice were provided to the experimental group, whereas booklets and person-to-person individual lectures were provided to the control group. Compliance with patient safety, the perception of patient safety culture, and satisfaction were measured using a structured Likert questionnaire. Intervention effects were analyzed using a *t*-test and ANCOVA. Results: As a result of the analysis, we found that the experimental group was significantly higher in terms of compliance with patient safety (*p* = 0.021), the perception of patient safety culture (*p* = 0.039), and education satisfaction (*p* < 0.001) than the control group. Conclusions: The results indicate that implementing a simulation-based patient safety education program can improve the patient safety competency of operating room nurses and, ultimately, prevent patient safety accidents in the operating room.

## 1. Introduction

With the advancement of science and technology, medical technology is also advancing. However, despite technological developments in the health system, medical errors (MEs) still continue to threaten patient safety (PS) in healthcare practices. Patient safety refers to the safe condition of being free from harm due to preventable events [1], and patient safety is an important indicator in the evaluation of the healthcare services provided by hospitals. Operating rooms have a high risk of patient safety incidents due to their complex structure and the noise from surgical tools, ventilation, and radiology equipment [2]. Patient safety incidents in the operating room include foreign-body residue in the surgical site after surgery, blood transfusion errors, and surgery in the wrong area [3], and these incidents can be life-threatening, so it is important for a medical team to maintain their current practices for patient safety.

Despite efforts to prevent patient safety incidents in the OR in South Korea, 272 of the 5202 patient safety incidents reported across all healthcare organizations in 2019 occurred in the OR, an increase of nearly 20% from 229 in 2018 [4]. Some of the reported patient safety incidents resulted in long-term damage, permanent damage, and death. Therefore, to reduce the number of patient safety incidents in the operating room, it is important to strengthen patient safety management activities and the role of operating room nurses who provide direct care [5].

Medical knowledge and technologies have developed rapidly. In particular, the operating room has changed and developed more than other departments in terms of facilities and roles [6]. Operating room nurses face a rapidly changing surgical environment, including new surgical instruments and equipment specialties [7]. The specifications and professionalism of medical services have lowered medical staff’s responsibility consciousness and safety consciousness, which can often cause patient safety accidents [8].

When looking at the status of medical disputes related to medical accidents, the surgical field, which includes 13 fields, such as diagnosis and examination, has the highest number of court cases at 36.9%, indicating that safety management in the surgical field is urgent, and patient safety is emerging as a major social concern [9]. In medically advanced countries such as the United States and the United Kingdom, an organizational culture has been formed in which medical errors can be openly discussed at the national level. In the United Kingdom, the government announced measures against medical errors and emphasized the importance of preventing medical errors that can lead to patient safety incidents through the National Patient Safety Agency [10].

Domestically, since the Korean Healthcare Accreditation System was introduced in 2004, interest in patient safety has become prevalent. With the improvement in medical consumers’ rights and the perception change in medical institutions, patient safety has emerged as an important issue [11]. The Ministry of Health and Welfare [12] incorporated patient safety factors into the Healthcare Accreditation System. As a result, medical institutions endeavored to improve patient safety and the quality of medical service by employing patient safety staff and organizing a patient safety committee.

The operating room features access control and complicated surgical operations in a closed environment, which can have dangerous effects related to patient safety accidents that can severely affect patients [13]. Additionally, patient accidents can cause psychological problems among medical staff. Both of these issues pose risks that can cause patient safety catastrophes, which can have severe consequences [14]. According to the World Health Organization [15], at least a million patients die during or immediately after a surgical operation. Of medical accidents related to safety rules, operating room accidents account for approximately 54% [16]. Cho [17] reported that 61 (24.6%) of the accidents related to patient safety in the operating room are caused by medical equipment malfunction; 53 (21.4%) are due to mismatched numbers of gauze pieces and needles; 34 (13.7%) are caused by specimen loss and specimen change; 15 (6%) are injury accidents caused by falls, burns, and pressure; and 11 (4.4%) are reoperation accidents caused by impurities in the body discovered by X-ray examination following a surgical procedure. This indicates the necessity for an organized and systematic management system to prevent operating room accidents.

To prevent patient safety accidents in the operating room, the current state of patient safety accidents and their relevant causes must be examined, and an education program must be developed and applied [18]. Furthermore, in addition to theoretical education, practical education on how to prevent and respond to accidents must be provided to improve the safety knowledge of operating room nurses [19]. However, most previous studies related to patient safety among operating room nurses focused on the perception, influencing factors, and performance of patient safety culture. Simulation training has the advantage of not being performed directly on patients, being safe, allowing for as much repetitive experience and learning as desired, and allowing students to experience cases that are difficult to perform in the clinic [20,21]. There were very few previous studies that provided simulation-based hands-on training to operating room nurses [22]. Simulation training is safe because it is not performed directly on patients, and it has the advantage of allowing for as much repetitive experience and learning as desired, as well as pre-experiencing cases that are difficult to perform in the clinic [20]. Simulation-based education has been shown to improve clinical performance and problem-solving processes compared to traditional lecture-based education or education utilizing videos [23,24]. Therefore, to improve the patient safety knowledge of operation room nurses, this study aimed to develop a simulation-based patient safety education program, define its effects, and propose an operating room patient safety education program.

The purpose of this study was to (1) develop a simulation-based patient safety education program for managing patient safety in the operating room and (2) to determine its effectiveness after applying it to operating room nurses and determine its effects on compliance with patient safety, the perception of patient safety culture, and education satisfaction.

**Hypothesis** **1**.
*The experimental group that participated in the simulation-based patient safety education program will have higher compliance with patient safety than the control group that did not participate.*


**Hypothesis** **2**.*The experimental group that participated in the simulation-based patient safety education program will have a higher perception of patient safety culture than the control group that did not participate*.

**Hypothesis** **3**.*The experimental group that participated in the simulation-based patient safety education program will have higher educational satisfaction than the control group that did not participate*.

## 2. Materials and Methods

### 2.1. Research Design

This study was an equivalent research design with a pretest–posttest control group to determine the effectiveness of a simulation-based patient safety education program for operating room nurses. To randomize the subjects, a die was rolled once the subjects were selected, and they were then assigned to the experimental group on an odd number and to the control group on an even number. To minimize treatment diffusion, the experimental sessions were separate so that the control group could not obtain experimental information from the experimental group. To enhance the validity of the study, double blinding was used to reduce the halo effect and separate the researchers into treatment providers and data collectors. The treatment providers were blinded to whether the subjects were in the experimental or control group. In addition, to reduce the Hawthorne effect in this study, subjects were blinded to whether they belonged to the experimental or control group.

### 2.2. Research Participants

This study’s participants were operating room nurses who voluntarily participated after seeing the recruitment notice through an online community. The participants were informed of the purpose and method of the study. The inclusion criteria for the participants were that the nurse is currently employed as an operating room nurse and conducts scrub and circulating nurse work in the operating room.

Based on the evidence that an effect size (ES) of about 0.56 was appropriate in a preceding meta-analysis study [25] on the effect of a simulation-based educational program to calculate the number of participants, the power (1 − ß) was set at 0.80, the significance level (α) for the two-tailed test was set at 0.05, the effect size (ES) was set at 0.50, and there were 34 cases when the ANCOVA analysis method was used. Therefore, considering the dropout rate, 50 cases were selected. The selected participants were assigned to the experimental group (22 persons) if an odd number was obtained by throwing some dice and to the control group (28 persons) if an even number was obtained. In the final analysis, 22 participants were in the experimental group, and 23 were in the control group. Five participants in the control group did not participate in the posttest due to medical leave or leaving the company.

### 2.3. Research Procedure

The research was conducted from 11 December 2017 to 28 February 2018. Each member of the experimental and control groups participated in two 60 min sessions of the program. Person-to-person education was also provided. Both groups were provided with the same educational content, number of educational sessions, and amount of time. The person-to-person individual lectures and simulation-based practice were provided to the experimental group, whereas person-to-person individual lectures were provided to the control group. The two educators of both groups were charge nurses in the surgical operating room with 15 years of operating room experience. Compliance with patient safety, the perception of patient safety culture, and educational satisfaction were measured using structured self-reported questionnaires before and immediately after the program began (Figure 1). 

### 2.4. Simulation-Based Patient Safety Education Program

#### 2.4.1. Program Development

The simulation-based patient safety education program was developed by a team comprising one leader of the operating room nurses, one education nurse from the operating room, one professor of nursing science who had a career in the operating room, and one doctoral student who had received simulation education. The operating room patient safety education program was established based on previous studies on patient safety [22,26], operating room nurses [25,27], and simulation education [25,28].

The first education session focused on patient safety and infection control, whereas the second focused on patient safety and operating room management. The method of education was simulation-based team education, the number of education sessions was 2, and the duration of each education session was 60 min, for a total of 120 min per team.

The content validity of the program was reviewed by two experts with over 10 years of work experience in operating rooms, one nursing professor with work experience in operating rooms, and one nursing professor with experience in developing and implementing patient safety-related education programs. Content validity was configured on a 4-point scale to prevent bias toward the midpoint score. The results confirmed that the CVI of the scenario content and checklist was 0.80 or higher.

#### 2.4.2. Program Progress and Contents

In this study, each person in the experimental group and the control group performed the sessions twice, 60 min each time, for a total of 120 min. The training was team-based, and the details of the training were as follows.

In this study, nurses in charge of surgical operating rooms with more than 15 years of operating room experience trained each team. The experimental group proceeded through the prebriefing, simulation, and debriefing stages, and the control group proceeded through the prebriefing and lecture-type education.

In the prebriefing stage, an orientation for patient safety education and education on patient safety culture awareness and implementation were conducted for 20 min for the experimental group and the control group.

The experimental group proceeded with simulation-based education using patient safety in the simulation stage. The contents of education were patient safety and infection control in the first session and patient safety and operating room management in the second session. Each session lasted 40 min and was conducted twice. In the debriefing stage, feedback was discussed interactively after patient-safety-based simulation education for 20 min. In contrast, for the control group, lecture-type education using the existing booklet and presentation program was conducted twice for 60 min per team (Table 1).

### 2.5. Measures

#### 2.5.1. General Characteristics

The general characteristics of the subjects in this study were measured using a questionnaire consisting of 12 items, including age, gender, marriage status, education, experience in the operating room, working hours, work type, position, department in the operating room, number of surgical patients, and experiences in patient safety education and simulation education.

#### 2.5.2. Compliance with Patient Safety 

Compliance with patient safety means practically preventing and managing what is recognized as important in hospitals [29]. The level of compliance was measured using a questionnaire first developed by Jang [30] based on the safety management guidelines of the Korea Association of Operating Room Nurses and Hospital Nurses Association and then modified by Kim and Kim [31]. The 5-point-scale-based questionnaire comprised 6 categories and 69 items: 8 items in the subcategory of specimen management, 12 items in the subcategory of infection control, 11 items in the subcategory of preoperative check, 15 items in the subcategory of medical equipment management, 8 items in the subcategory of damage prevention, and 14 items in the subcategory of count. The higher the number of points, the more safety management was ensured in the operating room. Regarding tool reliability, in Kim and Kim’s study [31], Cronbach’s α was 0.90, whereas in this study, Cronbach’s α was 0.94.

#### 2.5.3. Perception of Patient Safety Culture

In this study, the questionnaire used for safety culture perception analysis was a tool modified for the operating room based on the Safety Attitudes Questionnaire (SAQ) by Jang [32]. The five-point-scale-based questionnaire comprised 6 categories and 30 items: 5 items in the subcategory of job satisfaction, 6 items in the subcategory of organizational culture, 7 items in the subcategory of patient safety atmosphere, 4 items in the subcategory of working conditions, 4 items in the subcategory of hospital administration, and 4 items in the subcategory of stress. The higher the score, the more respondents were aware of safety. Regarding tool reliability, in the research by Jang [32], Cronbach’s α was 0.92, whereas in this study, Cronbach’s α was 0.97.

#### 2.5.4. Satisfaction with Education

Educational satisfaction refers to satisfaction with the program. Educational satisfaction was measured using the educational satisfaction questionnaire developed by Yoon [33] and modified by Lee and Kim [22]. The five-point-scale-based questionnaire contained 13 items. After excluding an item that was irrelevant to the jobs of the research participants (“the program helped to educate patients and their guardians”), 12 items were used. The higher the score, the more satisfied the participants were with their education. Regarding tool reliability, in the research by Lee and Kim [22], Cronbach’s α was 0.97, whereas in this study, Cronbach’s α was 0.95.

### 2.6. Analysis

The collected data were analyzed using SPSS Statistics 25.0, with the following statistical analysis methods:

(1)The general characteristics of the research participants and the homogeneity of research variables were analyzed using the chi-square test, Fisher’s exact test, and *t*-test(2)The effect of the simulation-based patient safety education program on operating room nurses was analyzed using ANCOVA, which homogeneously processed predefined values using a covariate. All statistical significance levels were *p* < 0.05

### 2.7. Ethical Considerations

For the ethical protection of research participants, this study was conducted only after obtaining approval from the Institutional Review Board of K Hospital, where the researcher was employed (IRB Approval No.: IRB- KUH 280085). Prior to data collection, informed consent was obtained from participants regarding the purpose and methodology of the study, the confidentiality and anonymity of their personal information, and their voluntary participation. Furthermore, there was no disadvantage in the event of a withdrawal from the study. The researcher explained that the collected data would be used exclusively for research and would be discarded three years after the conclusion of the study. All participants were given a gift as a reward for participating in the study, and willing control group participants were provided with the same program as the experimental group after the experiment was completed. 

## 3. Results

### 3.1. Homogeneity Test of General Characteristics

The participants were divided into two groups: 22 participants in the experimental group and 23 participants in the control group. No differences in age, operating room work career, work hours, sex, final education career, work type, position, department in the operating room, number of patients who underwent operations, experience with patient safety education, or experience with simulation education were identified between the two groups. Therefore, they were found to be homogeneous (Table 2). 

### 3.2. Effects of the Simulation-Based Patient Safety Education Program 

#### 3.2.1. Compliance with Patient Safety

Hypothesis 1. The experimental group that participated in the simulation-based patient safety education program will have a higher level of compliance with patient safety than the control group that did not participate. In this study, the score for patient safety implementation in the experimental group was 34.39 points, and that of the control group was 33.33 points; there was a statistically significant difference between the two groups (F = 6.80, *p* = 0.013). In each subdomain of compliance with patient safety, the experimental group was found to have higher scores than the control group; tissue specimen management (F = 4.44, *p* = 0.041), management of infectious patients (F = 5.94, *p* = 0.019), use of electrocautery (F = 4.60, *p* = 0.038), and management of medical equipment and use of laser light (F = 5.13, *p* = 0.029) showed statistically significant differences between the two groups, so hypothesis 1 was accepted (Table 3).

#### 3.2.2. Perception of Patient Safety Culture 

Hypothesis 2. The experimental group that participated in the simulation-based patient safety education program will have a higher perception of patient safety culture than the control group that did not participate. In this study, the perception of patient safety culture in the experimental group was 4.31, and that in the control group was 3.96, which was a statistically significant difference between the two groups, so Hypothesis 2 was accepted (F = 6.702.13, *p* = 0.039) (Table 3).

#### 3.2.3. Educational Satisfaction

Hypothesis 3. The experimental group that participated in the simulation-based patient safety education program will have higher educational satisfaction than the control group that did not participate. 

The educational satisfaction of the participants in this study was 4.47 for the experimental group and 3.85 for the control group, which was a statistically significant difference between the two groups, so Hypothesis 3 was accepted (F = 21.03, *p <* 0.001) (Table 3).

## 4. Discussion

This study developed a simulation-based patient safety education program and verified its effects, namely, compliance with patient safety and the perception of patient safety culture, on the improvement of operating room nurses’ abilities to conduct patient safety management. Based on these results, the following conclusions were drawn.

The simulation-based patient safety education program developed for this study combined simulation- and pamphlet-based lectures. The control group received traditional pamphlet-based lectures, whereas the experimental group participated in a simulation-based education program. In the previous study, only lectures or practice were conducted, whereas in this study, practice was conducted along with lectures. In a previous study that carried out some of the practices, practice was conducted in the lab, whereas this study was conducted in the operating room where surgery was actually performed.

A significant difference was identified between the two groups. This result is consistent with that of previous research on the application of operating room safety nursing to standardized patients [22], according to which compliance with patient safety increased. Compliance with patient safety increased in previous research and this study given that the participants had the opportunity to apply their acquired theoretical and practical knowledge. Tissue specimen management, management of infectious patients, use of electrocautery, management of medical equipment, and use of laser light received significantly more points than other subcategories of compliance with patient safety. The reason is that the research participants discussed the patient safety actions that they applied in simulation practice during the debriefing after practice, which allowed them to determine and correct their errors independently. The points for tissue specimen management did not significantly increase in Lee and Kim’s research [22] but did increase in this study. Unlike theoretical education, which focuses solely on theories, simulation-based education enables nurses to manage tissue specimens directly under the supervision of an educator. Consequently, their compliance with tissue specimen management increased. Regarding the subcategory of count, it increased in Lee and Kim’s study [22] but not in this study. The reason is that the experimental and control groups in this research scored high points (4.9 out of 5) in the count subcategory. In previous studies on simulation-based education for nurses [28,34,35], clinical performance ability increased. These results are similar to those obtained in the present study. In a previous study, clinical performance was assessed using a self-administered questionnaire, whereas this study assessed compliance with patient safety directly in a nursing setting, which is more meaningful. Applying the simulation-based operating room patient safety education program developed in this research to operating room nurses for patient safety education can improve their compliance with patient safety in the operating room, thereby preventing patient accidents. 

Regarding awareness of patient safety culture, the experimental group that participated in the simulation-based patient safety education program scored 4.32 points, whereas the control group scored 3.96 points; therefore, the two groups were significantly different. Research on simulation-based patient safety education for nurses and its effects is scant. Therefore, directly comparing these effects is difficult. In theoretical education, participants learned about the importance of patient safety, and in practical education, they observed problems that arose when patient safety activities were not performed. In the debriefing, the educator and learners discussed patient safety, increasing their awareness of patient safety culture.

In this study, nurses who participated in the simulation-based patient safety education program were more satisfied with the program than those who received traditional lecture-based patient safety education. Nursing education generally involves educator-oriented group lectures. This study provided learner-oriented individual simulation education. Thus, the participants’ educational satisfaction increased. This result is similar to that of Park and Kim [26], who used small-group discussions in video education and employed the danger prediction training/learning method, as well as the results of Lee and Kim [22], who used simulation-based education. Compared to theoretical education, simulation education can be practical in a real operating room setting. Through debriefing, nurses were able to identify their weaknesses, which increased their educational satisfaction.

Overall, applying the simulation-based patient safety education program developed in this study to the operating room can increase nurses’ compliance with patient safety in the operating room, enhance their awareness of patient safety culture, and prevent patient safety accidents. However, this study was conducted in a single university hospital, and there are limitations in generalizing the results of this study because compliance with patient safety was self-assessed. In addition, there were no significant differences in the three items that require long-term learning (preoperative check, preventing fall and skin damage, and gauze count), so we believe that these items need further continuous education. It is necessary to repeat the study in multiple university hospital operating rooms, and it is necessary to objectively evaluate compliance with patient safety through observation methods. 

Simulation has been emphasized as one of the strategies to prevent patient safety errors [36]. Adamson [37] stated that simulation education can induce behavioral changes in future real-world clinical situations through a response to the learning process and the achievement of learning outcomes, leading to ultimate educational effects such as improved patient safety. Based on the results of this study and previous studies, we propose simulation-based education that can reproduce clinical situations to improve nurses’ patient safety competence. However, this study has limitations in understanding the experiences of the study participants because the data were collected through surveys before and after the simulation-based education program. Therefore, to develop effective simulation education, a mixed-methods study that analyzes participants’ experiences along with a quantitative study on the effectiveness of simulation-based patient safety education is needed. In addition, to increase nurses’ participation in simulation-based patient safety education, it is necessary to use a mixture of high-fidelity simulators and standardized patients or to conduct simulation education using various educational methods, such as VR and AR.

## 5. Conclusions

This study aimed to develop a simulation-based patient safety education program and study its effects on improving the patient safety management abilities of operating room nurses. The results demonstrate that nurses’ compliance with patient safety and perceptions of patient safety culture significantly increased, and they were highly satisfied with the education program. This indicates that implementing a simulation-based patient safety education program can enhance the patient safety competency of operating room nurses, which can ultimately prevent patient accidents in the operating room. 

## Figures and Tables

**Figure 1 healthcare-11-02824-f001:**
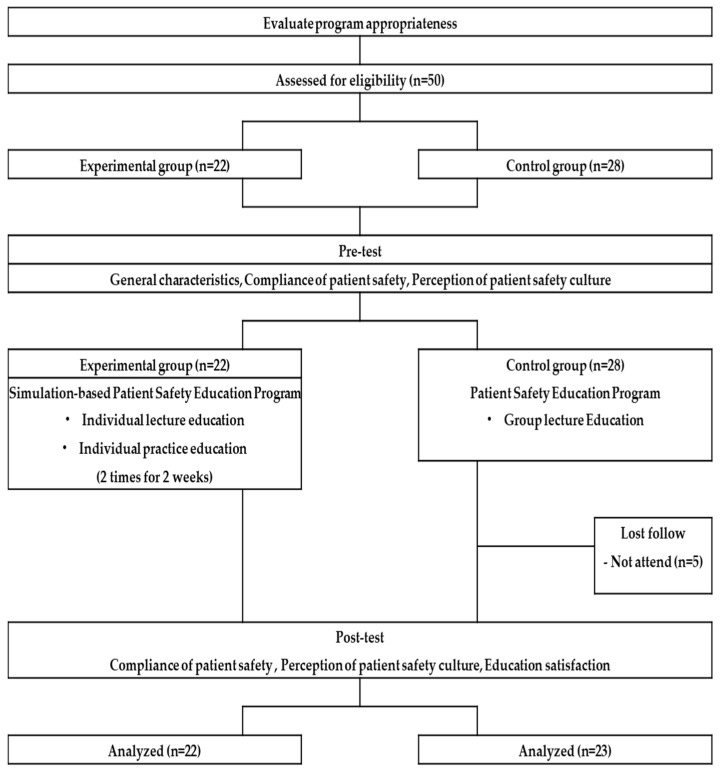
Research progress flow diagram.

**Table 1 healthcare-11-02824-t001:** Simulation-based patient safety education program for operating room nurses.

**Learning Goals**		
Participants can apply patient safety after the training program Participants can understand patient safety culture after the training program		
Session	Contents	Time (min)
Prebriefing stage	Orientation for patient safety education (operating room environment, safety rules, equipment, etc.) Awareness of patient safety culture Implementation of patient safety practices	20
Simulation stage	1. Patient Safety and Infection Control Infection prevention and management guidelines Aseptic technique and aseptic management PPE (Personal Protective Equipment) Management Specimen management	40 × 2 rate = 80
	2. Patient safety and operating room management Patient and operating room check (Time Out) Device handling and management (Equipment Count)	
Debriefing stage	Discussion of problem situations in simulation Discussion and suggestions for improvement of difficulties during the simulation process Evaluation and suggestions for improvement of analyzed nursing problems and plans Evaluation and suggestions for improvement of communication among patients and operating room nurses Feedback on points to improve during the simulation process	20

**Table 2 healthcare-11-02824-t002:** Homogeneity test of research participants’ general characteristics (N = 45).

General Characteristics		Experimental Group (n = 22)	Control Group (n = 23)	X^2^/t	*p*
		M ± SD /n (%)	M ± SD /n (%)		
Age (year)		35.82 ± 5.40	32.96 ± 5.34	1.79	0.081
Operation room experience (year)		11.26 ± 6.46	10.20 ± 5.72	0.58	0.563
Working time (h/wk)		42.77 ± 4.24	43.44 ± 2.73	−0.63	0.535
Gender	Women	19 (86.4)	17 (73.9)	1.09	0.297 *
	Men	3 (13.6)	6 (26.1)		
Education	Associate (3 years)	6 (27.3)	4 (17.4)	1.81	0.406 *
	Bachelor (4 years)	11 (50.0)	16 (69.6)		
	≥Graduate school	5 (22.7)	3 (13.0)		
Work type	3 shifts	16 (72.7)	15 (65.2)	0.30	0.749
	General shift	6 (27.3)	8 (34.8)		
Position	Clinical Nurse	9 (40.9)	8 (34.8)	0.18	0.672
	Nurse/Charge Nurse	13 (59.1)	15 (65.2)		
Department of operating room	Major part	12 (54.5)	13 (56.5)	0.02	0.991 *
	Minor part	4 (18.2)	4 (17.4)		
	Other	6 (27.3)	6 (26.1)		
Number of operations (n/day)	≤3	6 (27.3)	12 (52.2)	3.07	0.266
	4	6 (27.3)	5 (21.7)		
	≥5	10 (45.4)	6 (26.1)		
Experience with patient safety education	Yes	18 (81.8)	19 (82.6)	0.05	0.945 *
	No	4 (18.2)	4 (17.4)		
Experience with simulation education	Yes	7 (31.8)	7 (30.4)	0.10	0.920
	No	15 (68.2)	16 (69.6)		

* Fisher’s exact test.

**Table 3 healthcare-11-02824-t003:** Effects of the patient safety education program (N = 45).

Division	Categories	Exp.(n = 22)				Cont.(n = 23)				F *	*p*
		Pretest		Posttest		Pretest		Posttest			
		M ± SD	Interpretation	M ± SD	Interpretation	M ± SD	Interpretation	M ± SD	Interpretation		
Compliance with patient safety	Tissue specimen management	4.95 ± 0.14	Very high	4.95 ± 0.14	Very high	4.99 ± 0.04	Very high	4.81 ± 0.29	Very high	4.44	0.041
	Management of infectious patients	4.79 ± 0.36	Very high	4.71 ± 0.22	Very high	4.84 ± 0.28	Very high	4.45 ± 0.53	Very high	5.94	0.019
	Preoperative check	4.96 ± 1.22	Very high	4.95 ± 0.13	Very high	4.96 ± 0.11	Very high	4.87 ± 0.27	Very high	1.46	0.233
	Use of electrocautery	4.97 ± 0.15	Very high	4.98 ± 0.05	Very high	4.87 ± 0.27	Very high	4.78 ± 0.38	Very high	4.60	0.038
	Management of medical equipment and using laser light	4.89 ± 0.31	Very high	4.89 ± 0.13	Very high	4.91 ± 0.22	Very high	4.59 ± 0.70	Very high	5.13	0.029
	Preventing falls and skin damage	4.96 ± 0.08	Very high	4.93 ± 0.14	Very high	4.95 ± 0.10	Very high	4.85 ± 0.19	Very high	2.37	0.131
	Gauze count	5.00 ± 0.15	Very high	4.97 ± 0.04	Very high	4.98 ± 0.04	Very high	4.96 ± 0.07	Very high	0.240	0.625
	Total	34.51 ± 1.05		34.39 ± 0.54		34.57 ± 0.69		33.33 ± 1.97		6.80	0.013
Perception of patient safety culture		4.00 ± 0.62	High	4.31 ± 0.56	Very high	4.12 ± 0.59	High	3.96 ± 0.54	High	6.70	0.013
Educational satisfaction		4.06 ± 0.48	High	4.47 ± 0.42	Very high	4.13 ± 0.57	High	3.85 ± 0.69	High	21.03	<0.001

* ANCOVA results with a covariate pretest; Exp. = experimental group; Cont. = control group.

## Data Availability

Data supporting the findings of this study are available from the corresponding author upon request.

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
