# Peer review of "The Effects of a Simulation-Based Patient Safety Education Program on Compliance with Patient Safety, Perception of Patient Safety Culture, and Educational Satisfaction of Operating Room Nurses"

_healthcare, 2023, doi:10.3390/healthcare11212824_

Round 1
Reviewer 1 Report
Dear authors,
Your study is interesting and can be an asset to the scientific community, but the following points need to be clarified:
Methodology: what type of study is it: experimental study or which-experimental? is not well defined
Blind?
Did participants know they were in the experimental or control group? this information should be clear
it is necessary to better define the study design
Present in a joint table the pre-test and post-test results
Ask research questions or hypotheses

Author Response
We appreciate your review and will respond on file.

Reviewer 2 Report
Dear authors,
Thank you for the opportunity to review your work. Efforts to increase patient safety are important and I would like to applaud you for that Reviewing your work I have detected a number of issues.
Line 36 Why is patient safety a social issue. Could you please explain this.
Line 73 A previous study... was very poor. What do you mean by poor. Pour outcomes, poor study design etc
In designing your content for your sessions, how did you know that this content addressed the issues? Please explain.
Line 100 What is a post-mortem test?
Line 104 The data is from 2018 and why did it take 5 years to publish the results? The data is pre-covid and the world has changed in including healthcare. How do you know if your data still relevant in today's world?
Line 109 the word provided is used twice.
I am not convinced that you can change compliance by using two sessions., as this is a behavioral change occurring over a longer period. It is nice to know that participants are satisfied with the education provided, but this does change behaviour.
Line 238 I am concerned about providing participants with gifts, as this potentially leads to bias.
Table 2 Why is marital status important for this study? Are married people more compliant?
Line 301 -302 You are drawing strong conclusions, but this is not supported by the findings.
Overall I have concerns about the study design. Undertaking statistical analysis with 50 participants does not make sense. A more descriptive analysis would be more appropriate. You have not taken it consideration that people learn in different ways. You drawing strong conclusions based on two one hour sessions, using the same content but a different delivery methos. I would like to see more data so that the comparison becomes stronger.
The manuscript is well written with only a few grammatical issues.
Author Response

(The authors gave the same response as above.)

Reviewer 3 Report
Thank you for including me in this review. The study is interesting because it covers a topic that is of great magnitude and that is current. The study is well-designed and only some clarifications are needed, which I detail below.
Abstract
- Include a background that contextualizes the study.
- It is not relevant to detail the statistics used in the data analysis.
Introduction
- Do not use ; in line 30.
- Citation 4 is from 23 years ago, the data that reflect the magnitude of the problem are outdated, it would be appropriate to obtain a more recent source.
- It is necessary to restructure the introduction to include a description of the types of accidents related to patient safety and nursing, which are not sufficiently clear. On the other hand, the introduction does not contextualize the intervention, nor the educational programs that have been used most effectively.
Methods
- It is not necessary to include the study objective in the study design or to include the variables used.
- Do not repeat the study date, review the method and avoid duplications, they appear a lot.
- It is necessary to indicate how many simulation scenarios were carried out and what their specific learning objectives were. Was the simulation based on any quality standards?
- Table 1 has errors that need to be corrected.
Results
- Variables appear that are not detailed in the method, they should appear in the method.
Discussion
- It would be advisable to add a section on the limitations of the study.
Author Response

(The authors gave the same response as above.)

Reviewer 4 Report
Dear Authors,
Thank you for your interesting paper. It is highly relevant, however there are some points I suggest for you you to consider:
Title- The title should reflect the outcome being measured. What is the specific effect you want to measure from the operating room nurses?
Abstract
There are some grammatical issues that you need to address. Kindly consult an English language editor.
The second part of the aim of the study stated in the abstract implies that the program has been implemented. If not, then there is a need for your to clearly state the aim of your study. It is clear that you are developing the simulation -based patient safety education program and run them among the experimental group; however the second aim which states that: "to verify the effect of patient safety program."
Patient safety program is different from a simulation-based education program- kindly clarify. You also need to be specfic on what effect/outcome are you trying to measure.
It is not clear what kind of tool did you use to measure the outcomes- are these questionnaires? It was not stated.
You can revisit your abstract after considering my suggestions in the methodology and results section.
Introduction
The introduction can be improved by making an outline on how you should develop the discussion or presentation of your arguments.
You can better do this by clarifying first what is the real aim of your study:
Are you tring to test the effectivess of using simulation in educating the nurses about patient safety?
Are measuring the effectiveness of the simulation-based education program on the knowledge of the operating room nurses as mentioned in the last paragraph of your introduction, or are you measuring compliance, perception, and satisfaction of the OR nurses? Furthermore, the aim stated in the last paragraph of the introduction also stated development of patient safety education program. The aims in the abstract and in the introduction are not congruent. Kindly reconcile.
I suggest that you trim down the discussion on the prevalence of patient safety issues and adverse outcomes and add discussion on the following:
Simulation -based education - why did you opt for a simulation-based education more than other teaching/education strategies?
The thesis statement of your study and the support to this thesis statement.
Provide a strong link to the aim of the study (after youhave decided on what is really the aim of your study).
Discuss the outcomes ypu measured and why were these outcomes chosen. What is its relevance to pateint safety and to the simulation -based strategy of teaching.
State a strong aim of the study. It is the most important part of the introduction but it was given very limited discussion. There is a need for you to introduce the gap (global and local) that compelled you to come up with the simulayion-based education program.
State and support the hypothesis of your study.
Materials and methods
Design - You discussed again about your aim here and not the design. Kindly state what is the study design and why is this appropriate.
Participants - You can remove the inclusive dates as this was also included in the other section.
Interventions
The teaching method and content for the control group should also be described including the validity and reliability tests done.
Describe cases/ scenarios that were simulated and how the actual simulation was carried out during the education program.
Describe the simulation competence of the educator.
Controls- Include a section that discuss how you controlled intervening variables that may affect the outcomes.
Table 1 is out of placed.
Measures:
Based on the definition of compliance in your paper- it is best measured by observation on the actual practice. I suggest that you operatonally define the compliance as self- reported or perceived compliance since it was only determined by a questionnaire. May I know when did you measure the compliance? How many days after the SBE?
Perception- Is it perception or awareness that you measured?
When was satisfaction measure?
Where is the knowledge among your outcome variables? Your aim stated in the introduction part was on the knowledge of the OR nurses.
Results:
Table 2. Number of operation in a day - does it refer to the operations handled by the participant or the number of cases in the OR/day?
I suggest that you present and describe the levels of perceived competence and perception/awareness of patient safety culture from the pre test as well as the result of the post test . Then, the result from testing the significant difference within and between groups for both pre test and post test.
There is a need for you to measure the outcomes within the groups before comparimg between groups.
Inlude the decision on the hypothesis of the study.
There is a need to improve the presentation of the results of the study.
Discussion-
In you rdiscussion you mentioned pamphlet in addition to the simulation which was not included in the preceding parts of the paper. Be consistent.
State atb least the authors and year of the studies you arecreferring to in your discussion. It is not enough to state.. in the previous study... in another study...
What is the significant difference? Also it shoud be within group and btween groups.
Even before the study, it is expected that thereis significant difference on outcomes between simulation -based and traditonal methods of teaching, therefore you also focus on the features and requirements of a simulation-based education to justify the risks and benefits of using simulation. This could either be in the introduction part or in the methods section, or discussion part.
Kindly improve your conclusion on the merit f your aim and hypothesis.
Thank you and good luck.
There are grammatical errors across all sections that need to be addressed.
Author Response

(The authors gave the same response as above.)

Round 2
Reviewer 1 Report
Dear Authors,
tank you for the opportunity to review this paper.
The study is much better, more complete, clearer.
Only one question: In table 3, concerning the pre-test and post-test, is not clear if within the group there is an increase in knowledge or not.
Author Response
It has been edited based on the reviewer's advice and is attached as a file.
Q: Only one question: In table 3, concerning the pre-test and post-test, is not clear if within the group there is an increase in knowledge or not.
A:
|
Table 3 also describes the scores for the subdomains of Compliance of patient safety. The change in the total score of "Compliance of patient safety" remained unchanged from 34.51 to 34.39 in the experimental group, but decreased from 34.57 to 33.33 in the control group. This can be interpreted as a significant difference between the two groups in the ANCOVA analysis after controlling for the same prior values as covariates, indicating a relatively significant increase in the experimental group. |

Reviewer 2 Report
Dear authors,
Thank you for making the changes to your manuscript, it is a much improved version.
Line 61. The argument would be stronger if you incorporated the increased court cases as part of the social issue. This is a per your response questioning the social issue.
Table 2 the argument that martial status affects work performance does not stand. I would like to see this removed from the table.
Line 43 Sounds like you are arguing that medical professionals are currently not paying attention. This sentence needs to be rephrased.
line 46 Do you mean all healthcare organizations in Korea or globally.? Please clarify.
It remains unclear when participants took the survey, was it directly after the education session or much longer. Time lapsed would impact results. The result showed that compliance changed and this is measured by a self-reported survey. This is a limitation and should be acknowledge.
Some minor grammatical issues were detected and the manuscript good benefit from proof reading.
Author Response
It has been edited based on the reviewer's advice and is attached as a file.
|
Line 61. The argument would be stronger if you incorporated the increased court cases as part of the social issue. This is a per your response questioning the social issue. |
Since dispute means the number of court cases, we have modified it in the text. |
|
Table 2 the argument that martial status affects work performance does not stand. I would like to see this removed from the table. |
We have removed it based on the reviewer's comments. |
|
Line 43 Sounds like you are arguing that medical professionals are currently not paying attention. This sentence needs to be rephrased. |
and these incidents can be life-threatening, so it's important for a medical team to maintain their current practices for patient safety. |
|
line 46 Do you mean all healthcare organizations in Korea or globally.? Please clarify |
Despite efforts to prevent patient safety incidents in the OR in South Korea, |
|
It remains unclear when participants took the survey, was it directly after the education session or much longer. Time lapsed would impact results. The result showed that compliance changed and this is measured by a self-reported survey. This is a limitation and should be acknowledge. |
Compliance of patient safety, perception of patient safety culture, and educational satis-faction were measured using structured self-reported questionnaires before and immediately after the program began (Figure 1). |
|
Comments on the Quality of English Language Some minor grammatical issues were detected and the manuscript good benefit from proof reading. |
This manuscript has been formally proofread by a professional proofreader. We would appreciate any additional comments. (Relevant certificates attached) |

Reviewer 4 Report
Dear authors,
Thank you for your efforts in revising the paper. However one major area I pointed out was not clearly addressed.
I was looking for the pre and post test results for all variables with the interpretation. This descriptive part of the result will allow better appreciation for any change and difference within and between groups.
The table below is an example
|
|
Experimental |
Control |
||||||
|
outcomes |
Pre test |
Post test |
Pre Test |
Post test |
||||
|
|
Mean score |
Interpretation |
Mean Score |
Interpretation |
|
|
|
|
|
Satisfaction |
4.06 |
High |
4.40 |
Very high |
|
|
|
|
|
Compliance |
|
|
|
|
|
|
|
|
|
|
|
|
|
|
|
|
|
|
|
|
|
|
|
|
|
|
|
|
|
|
|
|
|
|
|
|
|
|
You as the researcher will develop the interpretation of the scores according to the scale used in your questionnaires and the intervals for each mean score.
example: for a 5-point likert scale, the interval is 0.8
therefore: 1.00- 1.80 can be interpreted as very low
1.81- 2.60 - Low
2.61-3.40- moderate.... etc
Furthermore, I suggest that compliance be described as intended compliance or self-reported compliance because compliance is not accurately measured by a questionnaire.
Was there an intervention done for the control group after the study? Like having them undergo thesame simulation -based training program received by the experimental group?
Thank you.
S
Minor editing for English language grammar is needed.
Author Response
It has been edited based on the reviewer's advice and is attached as a file.
|
I was looking for the pre and post test results for all variables with the interpretation. This descriptive part of the result will allow better appreciation for any change and difference within and between groups |
We added interpretations to Table 3 based on reviewer comments. 1.81- 2.60 - Low Ÿ 2.61-3.40- moderate Ÿ 3.41-4.20 high Ÿ 4.21-5.0.... vert high |
|
Furthermore, I suggest that compliance be described as intended compliance or self-reported compliance because compliance is not accurately measured by a questionnaire. |
Compliance of patient safety, perception of patient safety culture, and educational satis-faction were measured using structured self-reported questionnaires before and immedi-ately after the program began (Figure 1). |
|
Was there an intervention done for the control group after the study? Like having them undergo thesame simulation -based training program received by the experimental group? |
All participants were given a gift as a reward for participating in the study, and a willing control group was intervened with the same program as the experimental group after the experiment was completed. |
